# Identification and Characterization of a Mutant *pv-pur* Gene Responsible for the Purple Phenotype of Snap Bean (*Phaseolus vulgaris* L.)

**DOI:** 10.3390/ijms23031265

**Published:** 2022-01-23

**Authors:** Chang Liu, Xiaoxu Yang, Zhishan Yan, Dajun Liu, Guojun Feng

**Affiliations:** Academy of Morden Agriculture and Environmental Sciences, Heilongjiang University, Harbin 150000, China; lcky0909@126.com (C.L.); sunny19880106@126.com (X.Y.); 1994028@hlju.edu.cn (Z.Y.)

**Keywords:** snap bean (*Phaseolus vulgaris* L.), mutant, anthocyanin, transcriptome, metabonomics

## Abstract

Pod color is a major economic trait of snap beans (*Phaseolus vulgaris* L.), among which the pod with a purple stripe is more attractive to people. A stable purple mutant with purple stripes on the pods was obtained by artificial mutagenesis with the high generation snap bean inbred line ‘A18-1’. In order to reveal the genetic factors and pathways responsible for the purple appearance in snap bean, we performed transcriptome and metabolome analyses using the green stem and yellow pod cultivar ‘A18-1’ and its purple mutant ‘*pv-pur*’ via ^60^Co-γ radiation. Transcriptome analysis showed that three genes in the anthocyanin biosynthetic pathway were differentially expressed, among which the expression level of *F3′5′H* (*Phvul.006G018800*) was increased in the mutant ‘*pv-pur*’, while expression of *F3′H* (*Phvul.004G021200*) and *ANS* (*Phvul.002G152700*) was downregulated. Anthocyanin-targeted metabonomics analysis showed significant differences in the contents of 10 metabolites between the wild type and mutant plants. Combined analysis of transcriptome and metabolomics showed that one differential metabolite, delphinidin, was related to the differential expression of *Phvul.006G024700*, *Phvul.002G152700*, and *Phvul.006G018800*. Based on the levels of six anthocyanins in wild type and mutant plants, we speculative that the purple appearance of the mutant ‘*pv-pur*’ is caused by the increased expression of *F3′5′H* (*Phvul.006G018800*), the key enzyme in the transformation from dihydroflavanol (DHK) to dihydromyricetone (DHM) in the anthocyanin biosynthetic pathway. The results lay a foundation for further studies on the molecular mechanism of anthocyanin synthesis in snap bean, and provide a framework for breeding different colors of snap bean.

## 1. Introduction

There are rich and colorful colors in the plant world. The existence of these colors is due to the existence of a variety of plant pigments in the plant tissues, including chlorophyll, carotenoids, photosensitive pigments, flavonoids and some algal pigments. Flavonoids are widely distributed in higher plants; they are a group of water-soluble phenolic compounds which mainly exist in the vacuoles and play an important role in signal transmission and the stress resistance of plants [1]. Anthocyanins are one type of flavonoid. The variety and content of anthocyanins are the main factors that cause the color changes of fruits, flowers and vegetative tissues in different plants. Anthocyanins are also the main bioactive substances for plants to resist biotic and abiotic stresses such as low temperature, drought, strong light, and ultraviolet radiation [2,3].

Anthocyanins are derived from flavonoid biosynthetic pathways, which have been well studied in some model plants such as *Petunia hybrida*, snapdragon, *Arabidopsis thaliana*, and maize [4]. The first step in the biosynthetic of flavonoids is catalyzed by chalcone synthase (CHS), a member of the polyketide synthase (PKS) family. This is a necessary step in the synthesis of all flavonoids, including colored anthocyanins and some colorless flavonoids, such as flavonol. CHS usually cooperates with chalcone isomerase (CHI), flavanone 3-hydroxylase (*F3′H*) and flavonol synthase (FLS), which is a key branch of flavonol synthesis [2,5]. Flavonols provide UV protection, can form nectar vessels visible to insects, and can be used as the co-pigment of anthocyanins [6]. The synthesis of colorless anthocyanins catalyzed by dihydroflavonol 4-reductase (DFR) is a key regulatory step. Subsequently, under the action of anthocyanin synthase (ANS), chromophores are produced, and anthocyanins are glycosylated [7].

The mechanism of anthocyanin synthesis and regulation has been widely studied in Arabidopsis, *Petunia*, grape and other model plants, and two groups of genes are known, namely structural genes and regulatory genes [8,9,10]. Structural genes encode enzymes directly involved in anthocyanin biosynthetic, including phenylalanine ammonia lyase (PAL), 4-coumaric acid CoA ligase (4CL), chalcone sythetase, chalcone isomerase (CHS), flavanone 3-hydroxylase (F3H), flavonoid 3′-hydroxylase (*F3′H*), flavonoid 3′5′-hydroxylase (*F3′5′H*), dihydroflavonol 4-reductase (DFR) and anthocyanin synthase (ANS) [9,11]. *F3′H* and *F3′5′H* are important enzymes for the synthesis of different colorless basic anthocyanins, and they belong to the cytochrome P450 family. *F3′H* belongs to the CYP75B sub-family of cytochrome P450-dependent monooxygenase (P450) superfamily, which converts dihydroflavanol into colorless cyanidin anthocyanins. *F3′5′H* belongs to CYP75A, a sub-family of the P450 superfamily. It can convert DHK and dihydroquercetin (DHQ) into dihydromyricetin, which is the precursor of delphinidin [12]. *F3′H* is responsible for the synthesis of anthocyanins from cyanidin, while *F3′5′H* is responsible for the synthesis of anthocyanins from delphinidin. In plants, the ratio of *F3′5′H* to *F3′H* regulates the apparent color of plants; the higher the ratio of *F3′5′H*/*F3′H*, the more purple coloration the plants display [13]. For example, purple and blue varieties in roses are relatively scarce. This is because the petals of ordinary roses do not contain *F3′5′H*, and lack anthocyanins based on delphinidin. If *F3′5′H* and *DFR* genes are overexpressed in rose, delphinidin will be accumulated in petals, thus showing blue color [14].

Snap bean is a leguminous vegetable crop, with tender pods as the edible organ. It is an important vegetable crop with delicious taste and is rich in nutrition. It is cultivated all over the world. As its edible organ, pod color is a very important economic trait of snap bean. Among them, pods with a purple halo or purple stripes are more attractive. However, it is not clear what genes regulate anthocyanin synthesis to make pods show purple. This may be due to the great difference of genetic background between different snap bean varieties. Therefore, in this study, we applied an artificially created purple mutant of snap bean. Mutants are important for the study of plant functional genomics. Mutation of some important genes may cause dramatic changes in anthocyanin synthesis. Using mutants is an effective research strategy for studying the mechanism of anthocyanin synthesis in snap bean. ‘A18-1’ is an important dwarf snap bean ecotype. Its stems and leaves are green, its pods are yellow, and it is widely planted in northern China and deeply loved by consumers. Herein, we constructed a gamma radiation mutant library of ‘A18-1’ and obtained many mutants, including some with yellow leaf, dwarf stature, and leaf shrinkage phenotypes. Among the mutants was one with purple stems and veins, and pods with purple/red stripes. This means that the two materials used in this study are completely consistent in genetic background except for mutant genes, and the mutant genes are likely to be important regulatory genes in the anthocyanin synthesis network. In order to reveal the physiology and molecular mechanisms underlying the purple mutant, we investigated the genetics of the mutant traits, performed transcriptome and metabolome analyses of mutant and wild type plants, and determined their anthocyanin contents. The results revealed the key genes and metabolites causing the changes in appearance of the mutant. This will lay the foundation for exploring the molecular mechanism by which the mutant gene regulates anthocyanin biosynthetic in snap bean and guides its breeding and improvement.

## 2. Results

### 2.1. A Purple Mutant Was Observed in A ^60^Co-γ-Induced Mutant Bank

A snap bean mutant bank was constructed by exposing dry seeds of dwarf snap bean variety ‘A18-1’ to ^60^Co-γ radiation at 150 Gy. Compared with wild type plants, the mutant bank showed extensive phenotypic variation in leaf color, leaf shape, plant height, maturity, sterility, and other traits. One of the mutants exhibited a purplish phenotype.

In our experimental materials, wild type ‘A18-1’ is the dwarf snap bean variety. Its cotyledons, hypocotyls, stems and leaves are green, flowers are light pink, and pods are yellow. However, the leaves, hypocotyls, stems and veins of ‘*pv-pur*’ mutant were purple, the flowers were also purple, and purple red stripes appeared on the surface of pods (Figure 1).

### 2.2. Genetic Analysis of the Mutant ‘pv-pur’

In order to study the genetic behavior of the purple mutant, we used wild type ‘A18-1’ and purple mutant ‘*pv-pur*’ as parents to generate a six-generation cross population, and investigated the phenotype of each generation. The results are as shown in Table 1. All the F_1_ plants showed the same phenotype as ‘*pv-pur*’, which indicated that the purple phenotype of ‘*pv-pur*’ was dominant and controlled by nuclear genes. In the backcross generation, the segregation ratio of F_1_ × ‘A18-1’ was 1.17:1 (χ^2^ = 0.0535 < χ^2^_0.05_ = 3.84), which was consistent with the theoretical ratio of 1:1; all F_1_ × ‘*pv-pur*’ plants were the purple mutant phenotype. In the F_2_ population, the segregation ratio of wild type ‘A18-1’ and purple mutant ‘*pv-pur*’ phenotype was 1:2.95 (χ^2^ = 0.0535 < χ^2^_0.05_ = 3.84), which met the expected Mendelian segregation ratio of 1:3. Therefore, this result indicated that the purple mutant character of ‘*pv-pur*’ was controlled by a pair of dominant nuclear genes, and we named the gene *pv-pur*.

### 2.3. The Total Anthocyanin Content of ‘*pv-pur*’ Is Significantly Higher than That of ‘A18-1’

The total anthocyanin contents of hypocotyl, leaf, flower, and pod wall tissues of wild type ‘A18-1’ and mutant ‘*pv-pur*’ were determined by UV spectrophotometer. The total anthocyanin content in different parts of mutant ‘*pv-pur*’ varied from 3.31 mg/g to 10.831 mg/g, and the total anthocyanin content in pod walls was lowest (3.31 ± 0.44 mg/g), followed by leaves (5.37 ± 0.20 mg/g), flowers 10. 206 ± 368 mg/g, and the total anthocyanin content was highest in the hypocotyl (10.83 ± 0.77 mg/g). The content of total anthocyanins in hypocotyl, leaf, flower and pod wall of ‘*pv-pur*’ was significantly higher than that of wild type ‘A18-1’. Among them, the total anthocyanin content in hypocotyls of mutant ‘*pv-pur*’ differed the most from that of wild type ‘A18-1’, and was 5.81 times higher (Figure 2).

### 2.4. Illumina Sequencing, DEG Analysis, Funcyional Annotation and Classification

In this study, we constructed six independent sequencing libraries. After the quality control of the original sequencing data, 21952601, 26966989, 22764371, 21999131, 23371341, and 22957321 clean reads were obtained in six samples. The quality of sequencing was further assessed. In the sample CK1 (A18-1), we investigated the distribution of various base contents. The results showed that the content of A, T, G and C fluctuated only in the first 10 bases, following which the base contents were basically the same and tended to be stable. The results of the other five samples were similar to those of CK1, which is in accordance with the sequencing requirements, and could be sequenced normally.

All clean reads were aligned to the reference genome. In order to truly reflect the expression of transcripts, we normalized the number of mapped reads and the length of transcripts, and used FPKM (Fragments Per Kilobase per Million) as an indicator to measure the expression of transcripts or genes. In this study, 209 DEGs (differentially expressed genes) were identified (Appendix A). Compared with CK, 132 genes were up-regulated and 77 genes were down-regulated in mutant M. Therefore, the number of up-regulated genes in purple mutant ‘*pv-pur*’ was higher than that of down-regulated genes. At the same time, we found 12 specifically expressed genes (only expressed in CK or M) in 209 DEGs. Amongst them, three genes were specifically expressed in wild type ‘A18-1’ (CK), and nine genes were specifically expressed in ‘*pv-pur*’ (M).

In order to study the function of differentially expressed genes, we analyzed the enrichment of GO (Gene Ontology) terms and KEGG annotations for 209 DEGs of M versus CK. In GO enrichment analysis, 50 GO terms were enriched (Figure 3), including 16 for ‘cellular components’, 14 for ‘molecular functions’ and 20 for ‘biological processes’. Among them, ‘catalytic activity’ contained the most DEGs in the entry of ‘molecular function’. We speculated that the change of anthocyanin content in the purple mutant ‘*pv-pur*’ may be related to the catalytic process, compared with the wild type (Figure 3). We carried out KEGG (Kyoto Encyclopedia of Genes and Genomes) enrichment analysis of all differentially expressed genes, and the results showed that the DEGs are significantly enriched in 12 metabolic pathways (Appendix A). Up-regulated DEGs were enriched in 10 metabolic pathways, including flavonoid biosynthetic (Appendix A). Down-regulated DEGs were enriched in four metabolic pathways, including isoflavone biosynthetic (Appendix A).

### 2.5. Key Genes Involved in Anthocyanin Biosynthetic Revealed by Transcriptome Analysis

BLAST, GO, KO and KEGG were used to annotate the DEGs to screen for genes related to anthocyanin biosynthetic. Of these, 27 DEGs were involved in the biosynthetic and metabolism of anthocyanins, including 11 structural genes, 13 regulatory genes and three other genes involved in anthocyanin biosynthetic (Table 2). In order to verify the accuracy of the sequencing results, 21 genes related to anthocyanin biosynthetic were selected and verified by qRT-PCR. Among them, there were 12 structural genes and nine regulatory genes. The internal standard gene used was actin. Compared with the sequencing results, the expression trend seen in qRT-PCR was consistent, indicating that the sequencing results were accurate (Figure 4).

In the sequencing data, 61 genes participated in the anthocyanin synthesis pathway, including seven PAL genes, three C4H (cinnamate 4-hydroxylase) pazigenes, 10 4CL genes, 14 CHS genes, four CHI genes, one F3H gene, two *F3′H* genes, two *F3′5′H* genes, eight DFR genes, one ANS gene, three UGTs genes, one 5MAT (5-O-glucoside-6″-O-malonyltransferase) gene and five 3AT (anthocyanidin 3-O-glucoside-6″-O-acyltransferase) genes. Only three genes were differentially expressed, namely *F3′5′H* (*Phvul.006G018800*), *F3′H* (*Phvul.004G021200*), and *ANS* (*Phvul.002G152700*). The expression of *Phvul.006G018800* encoding *F3′5′H* was up-regulated in the mutant ‘*pv-pur*’, and the other two genes encoding *F3′H* and *ANS* were down-regulated in the mutant ‘*pv-pur*’. This meant that the conversion from DHK to DHQ could be reduced, while the conversion to DHM could be increased. It can therefore be inferred that delphinidin and malvidin in this metabolic branch of DHM will increase, while the content of cyanidin and peonidin in the other branch will decrease due to the decrease in DHQ (Figure 5).

### 2.6. Anthocyanin-Targeted Metabolomics Analysis

The same test material as that used for transcriptome analysis was selected for anthocyanin targeted metabolome analysis. A total of 12 known anthocyanin metabolites were detected in this experiment, and the corresponding information is shown in Table 3. In order to compare the differences in concentrations of each metabolite in different samples among the 12 metabolites, we corrected the mass spectral peaks of each metabolite detected in different samples to ensure the accuracy of quantitative analysis.

The metabolites with fold change ≥ 2 and *p*-value ≤ 0.5 were selected as the final differential metabolites. In this study, there were 10 metabolites that were significantly different, among which five metabolites that were significantly up-regulated in the purple mutant ‘*pv-pur*’; these were: malvidin 3-O-galactoside, mallow 3-O-glucoside, delphinidin 3-O-glucoside, malvin and petunidin 3-O-glucoside. Five metabolites were significantly down-regulated in the purple mutant ‘*pv-pur*’; these were: peonidin O-hexoside, cyanidin O-syringic acid, delphinidin, cyanin and peonidin-3-O-glucoside chloride (Table 4).

The KEGG database was used to annotate the different metabolites. The results showed that five of the 10 different metabolites could be compared with KEGG data. Among them, anthocyanin, malvacanthin 3-O-glucoside, anthocyanin 3-O-glucoside, anthocyanin and petunia 3-O-glucose are involved in anthocyanin biosynthetic (Table 5), while anthocyanin is also involved in flavonoid biosynthetic.

### 2.7. Joint Analysis of Transcriptome and Metabolome

In order to better understand the relationship between genes and metabolites, different genes and metabolites in the same group were mapped to the KEGG pathway at the same time. Based on the correlation analysis of different metabolites and genes, we can directly see the relationship between metabolites and DEGs by selecting DEGs and metabolites with a correlation greater than 0.8, according to the pathway. Finally, we obtained a metabolite pme0442 (delphinidin) which was related to three DEGs: *Phvul.006G024700 (SHT)**, Phvul.002G152700 (ANS)* and *Phvul.006G018800 (F3′5′H)* (Figure 6). It is speculated that the change in the expression of these three genes may be the cause of the purple phenotype of the mutant ‘*pv-pur*’.

### 2.8. Upregulation of F3′5′H Expression Leads to Increased Levels of Three Anthocyanins in the Mutant ‘pv-pur’, Resulting in a Purple Phenotype

In the above results, it is not exactly clear which changes in gene expression caused which changes in anthocyanin content. Thus, we determined the content of six different anthocyanins in different parts of mutant ‘*pv-pur*’ and wild type ‘A18-1’ plants. The content of six anthocyanins (delphinidin, cyanidin, petunia, geranium, peonidin, and malvidin) in hypocotyls, leaves, and flowers of wild type ‘A18-1’ and mutant ‘*pv-pur*’ plants were determined by HPLC. The results showed that the six anthocyanins were significantly differentially abundant in different parts of wild type and mutant plants (Figure 7). In the mutant ‘*pv-pur*’, delphinidin, petunia, and malvidin were significantly more abundant than in wild type plants. By contrast, levels of the other three anthocyanins in the mutant ‘*pv-pur*’ were significantly lower than in wild type plants. This indicates that there are three branches in the anthocyanin biosynthetic pathway, and the content of three anthocyanins produced by *F3′5′H* increased, while the content of three anthocyanins produced from the other two branches decreased. We speculate that the content changes may be due to gene mutation, leading to changes in the three branching steps after DHK. Therefore, based on our experimental results, we conclude that the purple phenotype of the mutant ‘*pv-pur*’ is caused by the significant increase in the levels of three anthocyanins (delphinidin, petunia, and malvidin) due to increased expression of *F3*’*5*’*H* (*Phvul.006G018800*).

### 2.9. Prediction and Sequence Alignment of Purple Mutant Gene pv-pur

Based on the results of transcriptome, metabolome, anthocyanin content in mutant ‘*pv-pur*’ and wild-type ‘A18-1’ tissues, the purple mutant gene *pv-pur* was predicted. According to the transcriptome sequencing results, we know that only three genes in the anthocyanin synthesis pathway are differentially expressed, they are Phvul.006G018800 (*F3′5′H*), Phvul.004G021200 (*F3′H*) and Phvul.002G152700 (ANS). In the results of joint analysis of transcriptome and metabolome, we found that the differential metabolism of metabolite pme0442 (delphinidin) was related to the differential expression of three genes, which were Phvul.006G024700 (SHT), Phvul.002G152700 (ANS) and Phvul.006G018800 (*F3′5′H*). Among them *F3′5′H* was up-regulated in mutant *‘pv-pur’*, and *ANS* was down-regulated. The anthocyanin content assay showed that the anthocyanin content in mutant ‘*pv-pur*’ increased, so we inferred that Phvul.006G018800 (*F3′5′H*) was the mutant gene that caused the purple phenotype of the mutant ‘*pv-pur*’.

The gene Phvul.006G018800 was cloned and sequenced in wild-type ‘A18-1’ and purple mutant ‘*pv-pur*’, respectively. After comparison, it was found that there were four SNPs in the sequence of mutant ‘*pv-pur*’, of which the first and fourth SNPs were nonsynonymous mutations (Figure 8). The SNP located at 493bp of the coding region is responsible for the substitution of a cysteine (Cys) with a serine (Ser). The SNP located at 864bp of the coding region is responsible for the substitution of a histidine (His) with a glutamine (Gln). Functional domain analysis showed that the protein contained a P450 domain, located at 36–498bp. The first SNP mutation is located in this domain. The first SNP mutation is located in this domain. The mutation from Cys to Ser may affect the spatial structure and protein function. This further confirms our conjecture.

## 3. Discussion

Plant color variation is often accompanied by pigment content, plastid development, and abnormal photosynthesis. The content and types of pigments endow plants with rich colors and coloring patterns. Among them, anthocyanins endow plants with purple, red, blue, and other colors [10]. Abnormal changes in anthocyanin content often affect the color of plant appearance. The six most common anthocyanins are cyanidin, delphinidin, malvidin, pelargonidin, peonidin, and petunidin [15]. Among them, pelargonidin gives plants and orange-red color, cyanidin is associated with purple, and delphinidin, malvidin, and petunidin give plants a blue-purple color [8,16]. Snap bean, the second most important legume vegetable after soybean, is widely planted and consumed all over the world [17]. There are great differences in the color of pods and plants of different kidney bean varieties. In one study on purple and green snap bean cultivars, malvidin was found to be the main anthocyanin causing purple pod skin, and three regulatory genes were concluded to play an important role in the transcriptional activation of anthocyanin synthesis genes in purple snap bean [18]. These researchers used two different cultivated varieties in their research, while in our research we used cultivated varieties without purple appearance and purple mutants generated from wild type plants as research materials, hence the final results will be different. The experimental material used in this study is a purple mutant artificially mutated by ^60^Co-γ treatment and its wild type. The particularity of this material is that it can ignore the differences caused by genetic background, so then the genes related to anthocyanin synthesis can be identified more quickly and accurately.

In our study, we found that total anthocyanins in all parts of the mutant ‘*pv-pur*’ were significantly elevated compared with those of wild type plants. Among the six common anthocyanin types, the contents of delphinidin, petunia, and malvidin in the mutant were significantly higher than in the wild type, and these three pigments are blue-purple in color, corresponding to the phenotypic color change of our mutant ‘*pv-pur*’. The contents of the other three anthocyanins in the mutant ‘*pv-pur*’ were lower than in the wild type, which indicates that there are issues in the process of transformation from DHK to various colored anthocyanins in the anthocyanin synthesis pathway in the mutant ‘*pv-pur*’. This led to an increase in the content of delphinidin, petunia, and malvidin, derived from one of the branches of DHK transformed by *F3′5′H*, while the products of cyanidin, geranium, and peonidin, which are transformed by DFR and *F3′H*, were decreased significantly in abundance. Combined with the results of transcriptome analysis, these results are consistent with our speculation that there were only three DEGs in the anthocyanin synthesis pathway, among which *F3′5′H* was significantly upregulated in the mutant ‘*pv-pur*’ and *F3′H* was significantly downregulated in the mutant ‘*pv-pur*’. The significant increase in *F3′5′H* expression in the purple mutant ‘*pv-pur*’ can effectively guide the biosynthetic of delphinidin, petunia, and malvidin. The significant decrease in *F3′H* expression in the purple mutant ‘*pv-pur*’ will reduce the levels of cyanidin and peonidin in this branch. DFR exhibits a distinct substrate specificity; only when *F3′H* and *F3′5′H* are both inhibited does the direction of the anthocyanin synthesis pathway move toward the biosynthetic direction of geranium and its derivatives [19,20]. Therefore, the high expression of *F3′5′H* in the mutant ‘*pv-pur*’ will result in DFR substrate specificity, becoming an important factor which may be the reason for the decrease in geranium and its derivatives in the mutant ‘*pv-pur*’.

*F3′5′H* belongs to the cytochrome P450 family and is one of the most important enzymes in the anthocyanin biosynthetic pathway [21]. Its main function is to convert DHK and DHQ into DHM. DHM is the precursor for the synthesis of delphinidin, and *F3′5′H* is an essential enzyme in this step [12]. *F3′5′H* gene was first cloned in eggplant, which was also the earliest *CYP* gene cloned in plants. It was found that the *F3′5′H* gene was located in the hypocotyl of eggplant seedlings and was induced by white light [22]. After that, a gene homologous with *F3′5′H* of eggplant was cloned from *Petunia hybrida* flowers. Subsequently, the *F3′5′H* gene was cloned and isolated from some other plants, such as *Catharanthus roseus*, *Campanula campestris*, rice, *Arabidopsis thaliana*, etc. [23,24,25,26]. In previous studies, we found that the expression of the *F3′5′H* gene was spatiotemporal, but its expression in different plants and different parts was also different. With the increase of white light intensity, the expression of *F3′5′H* gene increased. In the study of *Petunia hybrida*, the *F3′5′H* gene was not expressed in leaves, but expressed in flower buds, and the expression was highest at the middle development stage [27]. In addition, in immunohistochemical localization analysis, *F3′5′H* was found to be dominant in the phloem [28]. In our study, we believe that the phenotypic variation of the purple mutant ‘*pv-pur*’ was caused by upregulation of *Phvul.006G018800* encoding the *F3′5′H* enzyme. The observation of the purple mutant ‘*pv-pur*’ showed that the purple coloration appeared in the hypocotyl, stem, vein, petal and pod wall (Figure 1), with higher amounts in hypocotyl, stem and vein, which was consistent with the dominant expression of *F3′5′H* in the phloem. We conducted shading treatment on the purple mutant ‘*pv-pur*’ and found that the part with shading treatment by using tin foil paper had no purple phenotype, while the part without shading showed a purple phenotype (Figure 9). This shows that our mutant gene is also induced by light, which is consistent with the results in eggplant. In addition, we cloned and sequenced the gene *Phvul.006G018800* in ‘A18-1’ and its mutant ‘*pv-pur*’. One of the nonsynonymous SNPs is located in the P450 protein domain, resulting in a cysteine to serine mutation. Cysteine generally forms a disulfide bond to stabilize the protein configuration, and is generally not able to tolerate mutations. Therefore, the mutation is likely to cause changes in protein structure and protein function. Therefore, we believe that the purple mutant ‘*pv-pur*’ of snap bean is caused by upregulation of *Phvul.006G018800,* which encodes the *F3′5′H* enzyme.

## 4. Materials and Methods

### 4.1. Plant Materials

A snap bean mutant bank was constructed by exposing dry seeds of dwarf bean variety ‘A18-1’ to ^60^Co-*γ* radiations at 150 Gy. A total of 328 mutants were screened in the M2 generation, one of which was a purple mutant. The wild-type ‘A18-1’ and the purple mutant ‘*pv-pur*’ were cultivated in a cold shed in Heilongjiang University, Heilongjiang Province, China, in 2019. The hypocotyls were extracted at seedling stage and wrapped in tin foil. After being frozen in liquid nitrogen, the hypocotyls were stored in −80 °C freezers for metabolomic studies and transcriptome sequencing. After podding, the hypocotyls, stems, leaves, flowers and pods of the plants were taken and stored at −80 °C for RNA extraction and determination of anthocyanin content.

### 4.2. Genetic Analysis of the Purple Phenotype of Mutant Plants

‘A18-1’ and ‘*pv-pur*’ were used as parents to conduct a six-generation cross. The phenotypic characteristics of F1, BC1 and F_2_ generations were investigated, and the segregation ratio was determined by χ^2^ (Chi square) test.

### 4.3. Determination of Total Anthocyanins

A sample of approximately 0.5 g was weighed and put into a pre-cooled mortar, and 5 mL of extraction solution (80% methanol: 5% hydrochloric acid: 15% ultrapure water) was added at the same time. The sample was ground into a homogenate. The liquid was kept away from light, and placed in a 4 °C refrigerator for 12 h. The sample was then centrifuged at 10,732.8× *g* for 15 min, and the supernatant was collected after centrifugation. The absorbance values at 657 nm (A657) and 530 nm (A530) were determined by UV spectrophotometry. The total anthocyanin content was determined according to the equation: total anthocyanin content = (A530 − 0.15 × A637)/0.5 [29,30,31].

### 4.4. Determination of Six Anthocyanins

The contents of six anthocyanins including delphinidin, cyanidin, petunia, pelargonidin, peonidin, and malvidin were determined by HPLC. For anthocyanin extraction and HPLC analysis, refer to the agricultural industry standard of the People’s Republic of China (NY/t2640-2014).

### 4.5. Transcriptome Sequencing Analysis

‘A18-1’ and ‘*pv-pur*’ hypocotyls were used as experimental materials, and three biological repeats were set in each group. ‘CK’ represents wild type ‘A18-1’ and ‘M’ represents purple mutant ‘*pv-pur*’. After total RNA was extracted from the samples, the library was established and sequenced by Biomarker Technologies Co., Ltd. (Beijing, China). After sequencing, repeated base calling steps were performed to eliminate unwanted raw reads, including those with junctions, unknown bases greater than 10%, and low-quality reads. Then, TopHat was used to compare clean reads to reference sequences (https://phytozome.jgi.doe.gov/pz/portal.html, accessed on 17 May 2019) [32].

The Fragments Per Kilobase per Million (FPKM) method was used to estimate gene expression. The screening of differentially expressed genes (DEGs) was carried out according to an experimental method published previously [33]. The false discovery rate (FDR) of the DEGs between the two samples were selected to be less than or equal to 0.001 and the log_2_ Ratio was set to be greater than or equal to 1. Next, Gene Ontology (GO) and the Kyoto Encyclopedia of Genes and Genomes (KEGG) analyses were performed on DEGs. Go enrichment analysis first compares all DEGs in the GO database (http://www.geneontology.org/, accessed on 17 May 2019), and then calculates the number of genes for each GO term. All GO terms highly enriched in DEGs were compared with the genomic background again by hypergeometric test, and the corrected *p*-value was less than or equal to 0.05. GO enrichment analysis of DEGs show the main biological functions of DEGs. KEGG is a database related to major metabolic pathways. The significant enrichment analysis of pathway takes the KEGG pathway as the unit and uses hypergeometric analysis to find out the significantly enriched pathways in DEGs compared with the whole genome. The Q-value is less than or equal to 0.05.KEGG, and enrichment analysis determines the main metabolic and signal transduction pathways the DEGs are involved in.

### 4.6. Targeted Metabolomics Analysis

The experimental material was the same as the transcriptome sequencing material, and each group was repeated three times. LC-MS/MS analysis was performed by Biomarker Technologies Co., LTD (Beijing, China). Based on the self-built database of the sequencing company and public databases of metabolite information including MassBank (http://www.massbank.jp/, accessed on 12 July 2019), KNAPSAcK (http://kanaya.naist.jp/KNApSAcK, accessed on 12 July 2019), HMDB (http://www.hmdb.ca/, 12 July 2019), MoToDB (http://www.ab.wur.nl/moto/, accessed on 12 July 2019), and METLIN (http://metlin.scripps.edu/index.php, accessed on 12 July 2019), qualitative analysis of the primary and secondary mass spectral data was performed. In the qualitative analysis of some materials, the isotopic signals, repeated signals containing K^+^, Na^+^ and NH_4_^+^, as well as the repeated signals of fragment ions with higher molecular weight were removed. Metabolites were quantified by multiple reaction monitoring (MRM) analysis of triple quadrupole mass spectrometry. After obtaining the mass spectral data of metabolites from different samples, the peak area of all mass spectra peaks was integrated, and the peaks of the same metabolite in different samples were integrated and corrected. After the quality control analysis of the samples, the differential metabolites were screened. The fold change was calculated and *p*-value was obtained by the Wilcoxon rank sum test. As screening criteria, the metabolites with fold change ≥ 2 and fold change ≤ 0.5 were selected as the final differential metabolites. The fold change was the ratio of expression between the two samples (groups).

### 4.7. Combined Analysis of Metabolomics and Transcriptome

The Pearson correlation coefficient was used to calculate metabolomic and transcriptome data integration. For this purpose, the average of all biological repeats for each cultivar in the metabolomic data and the average value of each gene expressed in the transcriptome data center was calculated. The variation multiples of the purple mutant ‘*pv-pur*’ were calculated in the metabolomic and transcriptome data and compared with the wild type. Finally, the coefficients were calculated from the log_2_ difference multiple of each metabolite and the log_2_ difference multiple of each transcript using Microsoft Excel. The correlation corresponding to R2 > 0.9 was selected. The relationship between metabolome and transcriptome was visualized by using Cytoscape (version 2.8.2, Boston, MA, USA).

### 4.8. qRT–PCR Detection

Using a Roche fluorescence quantitative PCR instrument (Basel, Switzerland) and Tiangen fluorescence quantitative kit (Tiangen, Beijing, China), a 20 μL reaction system was used. The three-step reaction procedure was as follows: 95 °C, 5 min, followed by 40 cycles of 95 °C, 10 s; 58 °C, 1 min. According to the sequence of differentially expressed genes, Primer 6.0 was used to design gene-specific primers, and actin was used as an internal standard gene (Table 6). All reactions were repeated three times, and the expression of related genes was calculated by the 2^−ΔΔ^CT method [34].

### 4.9. Statistical Analysis

The average value of three repetitions was taken for all measurement indexes. The significance of difference was tested by IBM spss 23.0 (http://www.ibm.com/cnzh/products/spss-statistics, accessed on 10 December 2021).

### 4.10. Gene Cloning and Sequencing

Hiscript^®^ II reverse transcriptase (vazyme, Nanjing, China) was used to reverse transcribe the first strand of cDNA. PCR amplification was carried out with cDNA as template, primer (L: ATGGACACCTTGTTCCTTGTGA, R: TTAACTTGGCATGGTTGGTTG) and DNA polymerase was LA Taq (TaKaRa, Beijing, China). PCR procedure: 95 °C for 5 min; 30 cycles at 95 °C for 30 s, 58 °C for 30 s, 72 °C for 1 min; 72 °C 5 min; 4 save. PCR products were sent to the company for sequencing (Tsingke Biotechnology Co., Ltd. Beijing, China).

## 5. Conclusions

In this study, we examined the purple mutant of snap bean induced by artificial mutation. The results showed that the content of anthocyanins in the mutant ‘*pv-pur*’ was significantly higher than that in the wild type, indicating that the mutant gene was the key gene for anthocyanin synthesis. Genetic analysis showed that the purple character of the mutant ‘*pv-pur*’ was controlled by a single dominant nuclear gene, which was named *pv-pur*. Through combined transcriptome and metabonomic analysis, we demonstrated that increased expression of *Phvul.006G018800,* which encodes *F3′5′H*, is the reason for the purple mutant phenotype. The results of this study will not only provide a theoretical basis for the study of genes related to anthocyanin synthesis, but also the identification of targets for snap bean breeding programs.

## Figures and Tables

**Figure 1 ijms-23-01265-f001:**
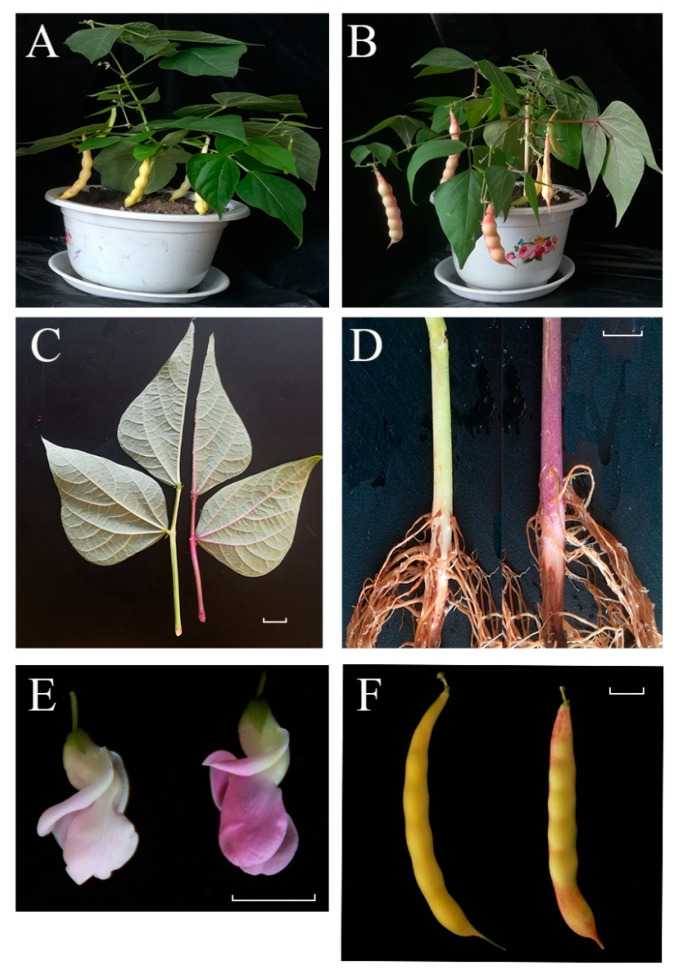
Phenotypic characterization of ‘*pv-pur*’ and wild-type ‘A18-1’. Notes: (**A**): wild type ‘A18-1’ in podding stage; (**B**): purple mutant ‘*pv-pur*’ plant in podding stage; (**C**): leaves and stems of wild type ‘A18-1’ (left) and purple mutant ‘*pv-pur*’ (right); (**D**): hypocotyls of wild type ‘A18-1’ (left) and purple mutant ‘*pv-pur*’ (right); (**E**): flowers of wild type ‘A18-1’ (left) and purple mutant ‘*pv-pur*’(right); (**F**): bean pods of wild type ‘A18-1’ (left) and purple mutant ‘*pv-pur*’ (right). The scale bar is 1 cm.

**Figure 2 ijms-23-01265-f002:**
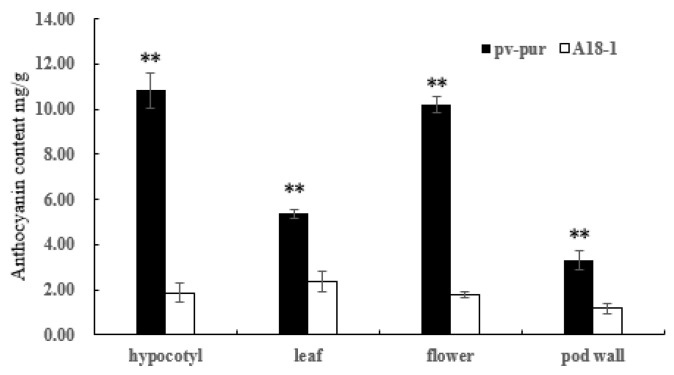
Anthocyanin content in various tissues of mutant ‘*pv-pur*’ and wild type ‘A18-1’. Note: ** represents significant differences at *p* ≤ 0.01 based on ANOVA (Tukey test).

**Figure 3 ijms-23-01265-f003:**
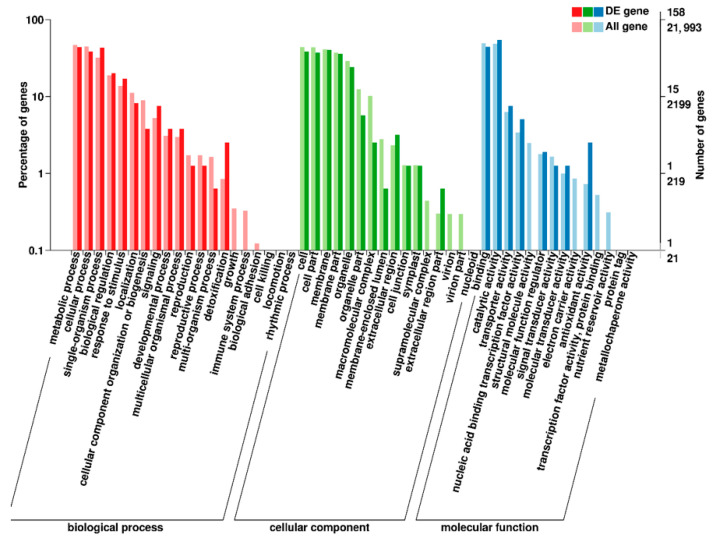
GO enrichment analysis in M versus CK. Notes: The abscissa is the GO classification, the left side of the ordinate is the percentage of gene number, and the right side is the gene number. M stands for mutant ‘*pv-pur*’ and CK stands for ‘A18-1’.

**Figure 4 ijms-23-01265-f004:**
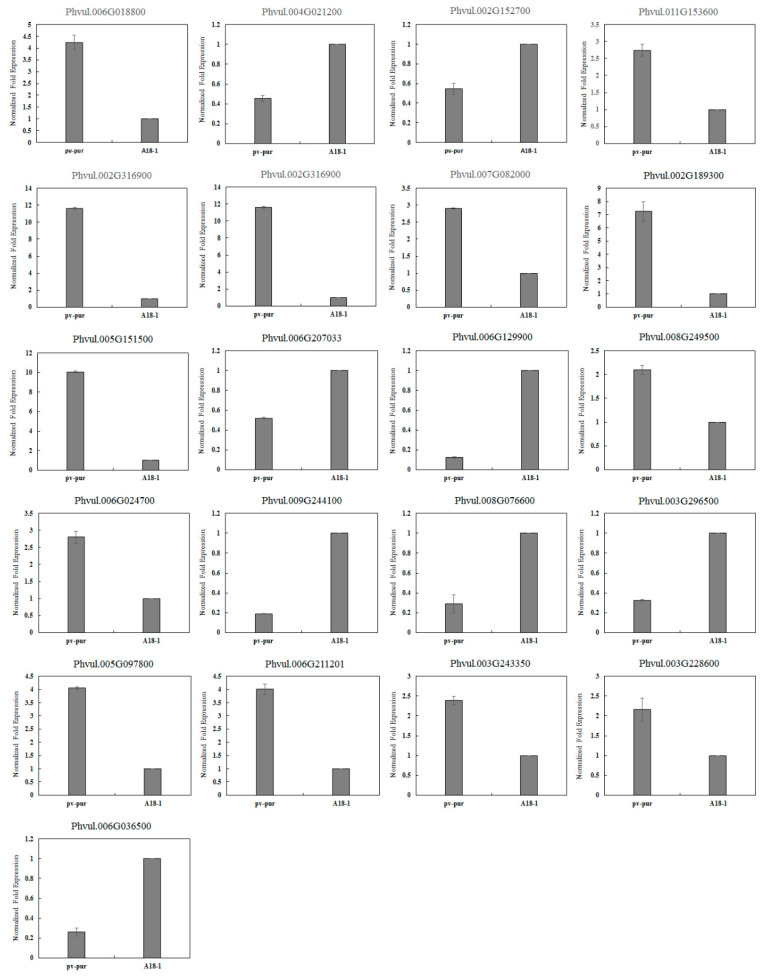
Validation of gene expression by qRT–PCR.

**Figure 5 ijms-23-01265-f005:**
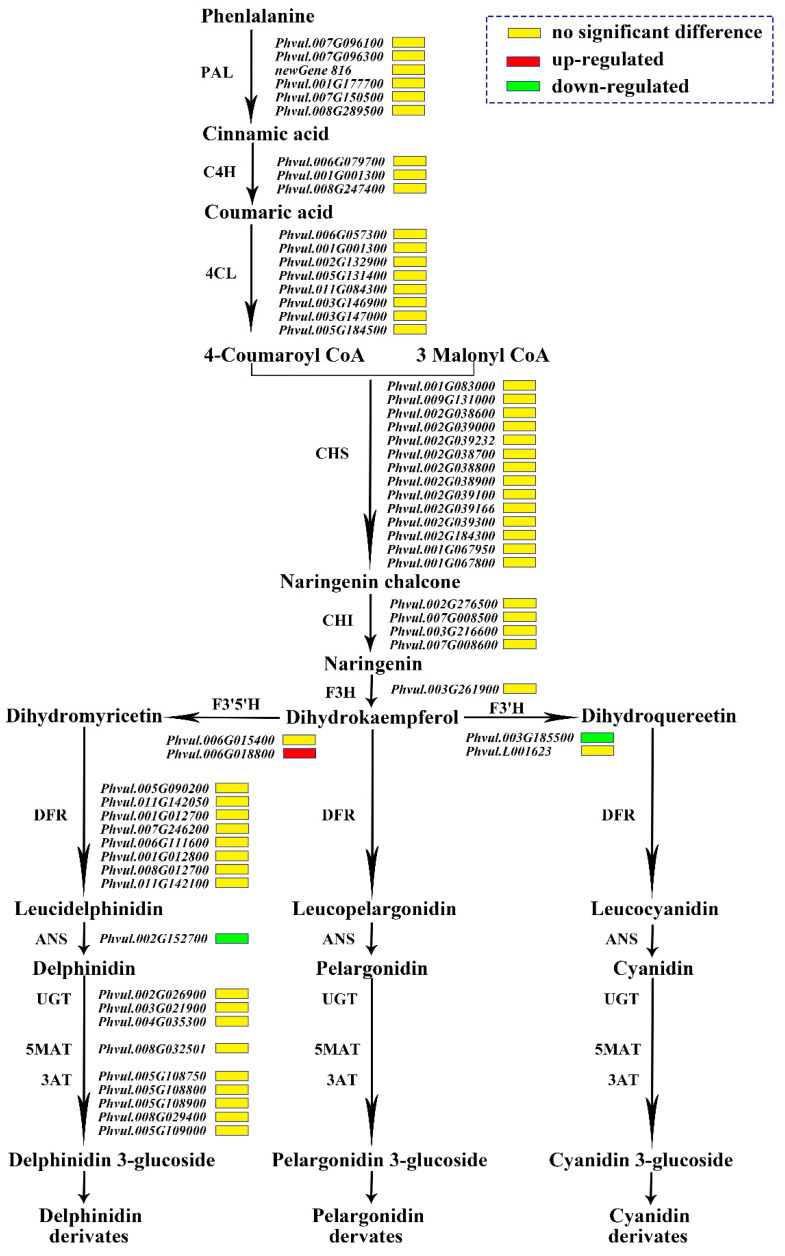
Expression patterns of the genes involved in anthocyanin biosynthesis. Notes: The name of the enzyme is marked on the left of the arrow, and the gene encoding this enzyme is marked on the right of the arrow. The red box represents that the gene is up-regulated in expression in ‘*pv-pur*’, while the green box represents down-regulated expression, and the yellow box represents no significant difference.

**Figure 6 ijms-23-01265-f006:**
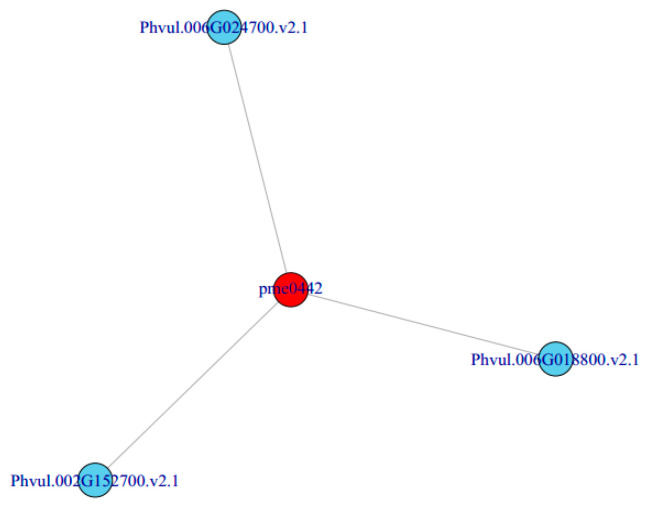
Correlation network of metabolites and genes.

**Figure 7 ijms-23-01265-f007:**
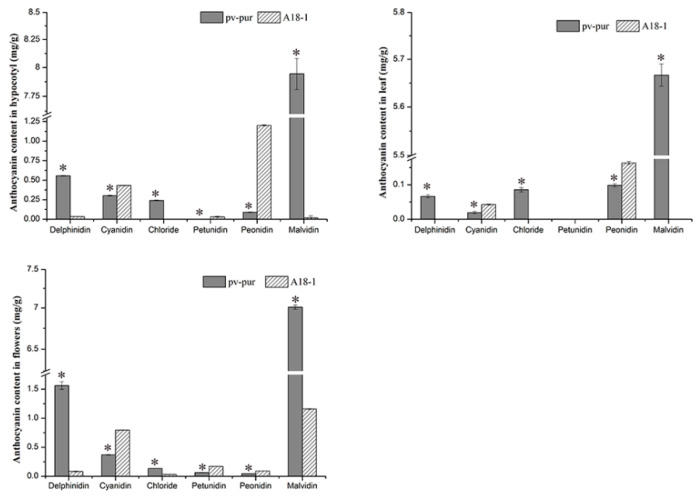
Concentration of six anthocyanins in different tissues of wild type ‘A18-1’ and mutant ‘*pv-pur*’. Note: * represents significant differences at *p* ≤ 0.05 based on ANOVA (Tukey test).

**Figure 8 ijms-23-01265-f008:**
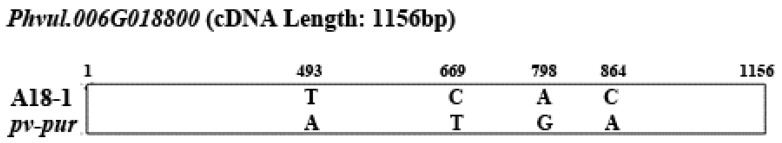
Sequence alignment diagram of *Phvul.006G018800* in wild type ‘A18-1’ and purple mutant ‘*pv-pur*’.

**Figure 9 ijms-23-01265-f009:**
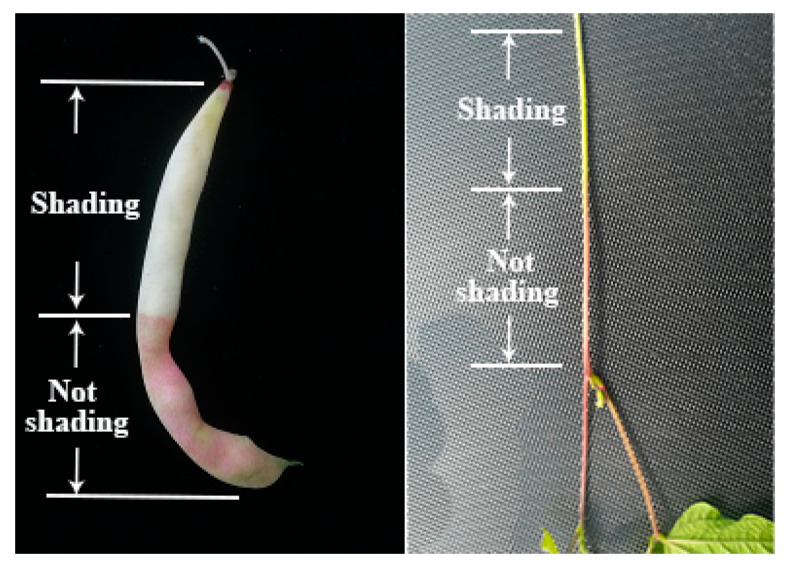
Pod and stem phenotype of ‘*pv-pur*’ after shading treatment.

**Table 1 ijms-23-01265-t001:** Genetic analysis of purple mutant *pv-pur* in *Phaseolus vulgaris*.

Generation	Total	A18-1	*pv-pur*	Segregation Ratio	χ^2^
P_1_ (A18-1)	25	25	0		
P_2_ (*pv-pur*)	25	0	25		
F_1_ (P1 × P2)	53	0	53		
F_1_ (P2 × P1)	47	0	53		
BC_1_ (F_1_ × ‘A18-1’)	39	21	18	1.17: 1	0.1154
BC_1_ (F_1_ × ‘*pv-pur*’)	40	0	40		
F_2_	1351	342	1009	1: 2.95	0.0158

Note: ‘Total’ represents the total number of plants, ‘A18-1’ represents the number of plants consistent with the wild-type phenotype, ‘*pv-pur*’ represents the number of plants consistent with the purple mutant phenotype.

**Table 2 ijms-23-01265-t002:** Differentially expressed genes involved in anthocyanin biosynthesis.

Classification	Gene ID	Ko	Log_2_ fc
**Synthetic gene**			
*F3′5′H*	*Phvul.006G018800*	K13083	8.69
*F3′H*	*Phvul.004G021200*	K00512	−2.54
*ANS*	*Phvul.002G152700*	k05277	−1.22
**Degradation gene**			
*POD*	*Phvul.008G249500*	k00430	1.61
	*Phvul.006G207033*	k00430	−2.14
	*Phvul.006G129900*	k00430	−2.27
*BGLU12*	*Phvul.005G151500*	k01188	3.71
*PRDX6*	*Phvul.002G189300*	K11188	4.35
**Phenylpropanoid**			
*SHT*	*Phvul.006G024700*	k13065	1.19
*VR*	*Phvul.008G076600*	k13265	−1.74
*I2’H*	*Phvul.009G244100*	K13260	−1.35
**Regulatory genes**			
*MTR_3g055120*	*Phvul.011G153600*		1.71
*HSFB3*	*Phvul.010G132433*		Inf
*bHLH153*	*Phvul.003G296500*		−1.25
*BEE3*	*Phvul.002G316900*		3.46
*HEC1*	*Phvul.003G243350*		3.44
*ARPC1B*	*Phvul.006G036500*	k05757	−2.70
*APRR2*	*Phvul.003G228600*		1.04
*GBF4*	*Phvul.006G211201*		2.64
*ERF110*	*Phvul.007G082000*		4.35
*RAP2−1*	*Phvul.001G023700*		1.01
*BZIP61*	*Phvul.005G097800*		3.35
*HSFB3*	*Phvul.007G251900*		−3.67
*AGL8*	*Phvul.009G203400*		−3.17
**Others**			
*CYP71D10*	*Phvul.006G209700*		3.56
	*Phvul.006G209500*		−2.28
	*Phvul.006G209600*		−4.17

**Table 3 ijms-23-01265-t003:** Statistical table of metabolite information.

Index	Compounds	Class	KEGG ID
pma1590	Peonidin O-hexoside	Anthocyanins	-
pmb0545	Rosinidin O-hexoside	Anthocyanins	-
pmb2957	Cyanidin O-syringic acid	Anthocyanins	-
pme0442	Delphinidin	Anthocyanins	C05908
pme0443	Malvidin 3-O-galactoside	Anthocyanins	-
pme0444	Malvidin 3-O-glucoside (Oenin)	Anthocyanins	C12140
pme1398	Delphinidin 3-O-glucoside (Mirtillin)	Anthocyanins	C12138
pme1777	“Cyanidin 3,5-O-diglucoside (Cyanin)”	Anthocyanins	C08639
pme1786	“Malvidin 3,5-diglucoside (Malvin)”	Anthocyanins	C08718
pme3391	Petunidin 3-O-glucoside	Anthocyanins	C12139
pme3609	Cyanidin	Anthocyanins	C05905
pmf0203	Peonidin 3-O-glucoside chloride	Anthocyanins	-

**Table 4 ijms-23-01265-t004:** Screening results of differential metabolites.

Compounds	*p*-Value	Log_2_ fc	Down/Up-Regulated
Peonidin O-hexoside	0.1	−3.997763584	Down
Cyanidin O-syringic acid	0.1	−3.525739329	Down
Delphinidin	0.1	−1.659700902	Down
Malvidin 3-O-galactoside	0.1	9.953751834	Up
Malvidin 3-O-glucoside (Oenin)	0.1	10.2034867	Up
Delphinidin 3-O-glucoside (Mirtillin)	0.06360257	19.86383696	Up
Cyanidin 3,5-O-diglucoside (Cyanin)	0.1	−3.916313106	Down
Malvidin 3,5-diglucoside (Malvin)	0.1	5.370178769	Up
Petunidin 3-O-glucoside	0.1	5.974442025	Up
Peonidin 3-O-glucoside chloride	0.1	−4.031065859	Down

**Table 5 ijms-23-01265-t005:** Annotation of differential metabolites in KEGG.

Metabolite	Compound ID	KEGG Pathway
Peonidin O-hexoside	-	-
Cyanidin O-syringic acid	-	-
Delphinidin	C05908	Anthocyanin biosynthesis; Flavonoid biosynthesis
Malvidin 3-O-galactoside	-	-
Malvidin 3-O-glucoside (Oenin)	C12140	Anthocyanin biosynthesis
Delphinidin 3-O-glucoside (Mirtillin)	C12138	Anthocyanin biosynthesis
Cyanidin 3,5-O-diglucoside (Cyanin)	C08639	Anthocyanin biosynthesis
Malvidin 3,5-diglucoside (Malvin)	C08718	-
Petunidin 3-O-glucoside	C12139	Anthocyanin biosynthesis
Peonidin 3-O-glucoside chloride	-	-

**Table 6 ijms-23-01265-t006:** Primers used for qRT-PCR.

Genes	Primer Sequence (5′-3′)
*RAP2-1 (Phvul.001G023700)*	F: TTCAACCATCACCAACAGA R: CCACTTCCTCATCCGTATT
*ANS (Phvul.002G152700)*	F: GAGAAGGAAGTTGGTGGAA
	R: GAGGAGGAAGGTGAGTGA
*PRDX6 (Phvul.002G189300)*	F: GGTGCCAAGGTGAATTATC
	R: GTTGCCAGTGGAGTCTT
*BEE3 (Phvul.002G316900)*	F: CCAGTGTTAGTTCCTATCAGT
	R: TCTCTTGCCTCTTCCAGAA
*APRR2 (Phvul.003G228600)*	F: GAGGTGAGTTCAAGCAGTA
	R: TGTGTTCAGGCAATGGTT
*HEC1 (Phvul.003G243350)*	F: AGGATAAGCGAGAAGATAAGG
	R: CGACAGCACCAACAGTAT
*bHLH153 (Phvul.003G296500)*	F: TAGAGGACGCAATGGAGTA
	R: GACACAGGAACAAGGCATA
*F3′H (Phvul.004G021200)*	F: ACTCTTGAATGCCTCACAA
	R: TGACACCGAACTTGATGG
*bZIP61 (Phvul.005G097800)*	F: GCCATTATCAGCATCATCAA
	R: GTTCTCACCACACCTTATTG
*BGLU12 (Phvul.005G151500)*	F: GCTTGCCAATGGTCTACA
	R: CAATGCGAGGACTTAGGAA
*F3′5′H (Phvul.006G018800)*	F: GAACAACAAGACGCTCATC
	R: TCTGCCAAGGACCACTC
*SHT (Phvul.006G024700)*	F: GTAAGGCTCGTGGATTAGAT
	R: TGTGATGATGCTGCTGTTA
*ARPC1B (Phvul.006G036500)*	F: GTGCCAACTCTTGTTATCC
	R: CGGTTATTATCCTGCTCATAG
*POD (Phvul.006G129900)*	F: TCAAGACAGCGGTGGAA
	R: AGTTGAGTGAGGTTGAAGAA
*PNC2 (Phvul.006G207033)*	F: TGCCTGGTCCTAATGATAAC
	R: GCGGTTGTGCCTATTGTA
*GBF4 (Phvul.006G211201)*	F: ATGTGCCATCTCAGAGTTC
	R: CAACGGAGGTCAACAACA
*ERF110 (Phvul.007G082000)*	F: CGAGCAGTGGTTCTATGAT
	R: AAGGAAGCAGAGGATGGT
*VR (Phvul.008G076600)*	F: AGTGCTTGGAGTGATGTG
	R: ACTGCCTTCTCTGTCAAC
*GSVIVT00023967001 (Phvul.008G249500)*	F: TGGTTGCGATGGTTCAG
	R: TGCTTCCACCTTAGACTTG
*I2’H (Phvul.009G244100)*	F: CTCGTCTCGTGGTTGTG
	R: CGGTGGTGTTGTCGTAG
*MTR_3g055120 (Phvul.011G153600)*	F: GCAATGGAGGAGGAGAAG
	R: GATGTCTTGGTAGGCTTGA
Actin	F: GAAGTTCTCTTCCAACCATCC
	R: TTTCCTTGCTCATTCTGTCCG

## Data Availability

Not applicable.

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
