# Peer review of "Identification and Characterization of a Mutant PV-PUR Gene Responsible for the Purple Phenotype of Snap Bean (Phaseolus vulgaris L.)"

_ijms, 2022, doi:10.3390/ijms23031265_

Round 1
Reviewer 1 Report
Liu et al. in their research article entitled ‘Identification and Characterization of a Mutant PV-PUR Gene 2 Responsible for the Purple Phenotype of Snap Bean (Phaseolus 3 vulgaris L.)’ conducted extensive omics analyses using transcriptome and metabolome tools in the green stem and yellow pod cultivar ‘A18-1’ and its purple mutant ‘pv-pur’ via 60Co-γ radiation. Authors indicated some structural genes, regulatory genes and different metabolites involved in anthocyanin biosynthetic transcriptome analysis and anthocyanin-targeted metabolomics analysis, and obtained a metabolite pme0442 (delphinidin) which was related to three DEGs: Phvul.006G024700 (SHT), Phvul.002G152700 (ANS) and Phvul.006G018800 (F3'5'H) by joint analysis of transcriptome and metabolome. Combined HPLC analysis indicted that the purple phenotype of the mutant is caused by the significant increase in the levels of three anthocyanins (delphinidin, petunia, and malvidin) due to increased expression of F3'5'H (Phvul.006G018800).I think the the original design of these experiments were trying to identified and found some links of the anthocyanin biosynthetic pathway genes that the changes in the expression of these genes may be the cause of the purple phenotype of the snap beanand mutant. However, authors didn’t follow this main idea at all, especially not in the Introduction and Discussion section to point out the particularity and individuality of genetic background of the materials. It needs some clarification and improvement in some points. In addition, please authors add the scale bar for all pictures in the text.
Overall, the data presented here would be pretty interesting for the readers of this journal.
Author Response
Dear Reviewer,
We have studied the valuable comments from you carefully, and tried our best to revise the manuscript. The point to point responds to the reviewer’s comments are listed as following:
Comment 1:Authors didn’t follow this main idea at all, especially not in the Introduction and Discussion section to point out the particularity and individuality of genetic background of the materials. It needs some clarification and improvement in some points.
Response: Thank you for your valuable advice. In the Introduction and Discussion, we added a description of the genetic background specificity of mutant materials and marked them with red revision pattern.
Comment 2:Please authors add the scale bar for all pictures in the text.
Response: Thank you for your careful work. We added a scale bar in Figure 1 and added notes.

Reviewer 2 Report
Liu et al reported the identification and characterization of a mutant strain of snap bean, named PV-PUR, through analyses of RNA-Seq and metabolome.
Those data are clear, however, they need to show additional evidence to support the authors' conclusion, and also should improve basic points before evaluation by reviewers.
I hope the following comments will help them to improve the article.
1) Abstract:
They started with " In order to .....". Before that, they need to mention the background of this study briefly.
2) The authors stated that the mutant PV-PUR is caused by an increment of F3'5'H expression. If so, they should show genetic and/or biochemical evidence to support the conclusion. At this stage, the conclusion of this study is just speculation.
3) Some Figures and Tables are lacking critical information:
The figure legends are so too bad to understand for the readers since there is no information explaining the data shown in each Figure.
In the case of Tables, it are lacking some important information and it are hard to understand. For example, Table 1, I could not understand the meaning of the numbers, and the meaning of "Total" is what? Table 2 must be shown the expression ratio at least.
4) some titles are not informative,
for example, "2.4 Transcriptome analysis" is lacking what you want to mention in the section by transcriptome analysis.
5) Figure 4:
Most of the genes' expression shown in Figure 4 was not found in Figure 5. Why you can judge whether the transcriptome data is accurate?
6) "mutant"
They used "mutant" in this study in many places.
They should clearly indicate "mutant" (PV-PUR?) since the readers are unable to understand which mutant they are saying.
7)A working model based on this study will give us additional information to understand this study and it also becomes more clear your conclusion.
8) The authors must clear the method for the "enrichment of GO terms and KEGG annotations for 209 DEGs".
9) relate to comment No. 6, CK1 and CK are not clear for what and which strain.
Author Response
Dear Reviewer,
We have studied the valuable comments from you carefully, and tried our best to revise the manuscript. The point to point responds to the reviewer’s comments are listed as following:
Comment 1:Abstract: They started with " In order to .....". Before that, they need to mention the background of this study briefly.
Response: Thank you for your valuable work. We added the background of the article at the beginning of the Abstract and marked it with red revision mode.
Comment 2:The authors stated that the mutant PV-PUR is caused by an increment of F3'5'H expression. If so, they should show genetic and/or biochemical evidence to support the conclusion. At this stage, the conclusion of this study is just speculation.
Response: Yes, what the reviewer said is quite right. The conclusion of this paper is only a speculation and is not verified by experiments. There is a problem with our formulation in the Abstract. Now we change it to ‘speculate’ on line 24.
Comment 3:Some Figures and Tables are lacking critical information:
The figure legends are so too bad to understand for the readers since there is no information explaining the data shown in each Figure.
In the case of Tables, it are lacking some important information and it are hard to understand. For example, Table 1, I could not understand the meaning of the numbers, and the meaning of "Total" is what? Table 2 must be shown the expression ratio at least.
Response: Thank you for your careful work. We carefully studied the Figure and Table and found that there were some incomprehensible places. We added notes to some Figures and Tables, and added the differential expression ratio of DEGs in Table 2. These changes are represented in red revision mode.
Comment 4:some titles are not informative, for example, "2.4 Transcriptome analysis" is lacking what you want to mention in the section by transcriptome analysis.
Response: We changed the subtitle of 2.4, and marked it with red revision mode.
Comment 5:Figure 4:Most of the genes' expression shown in Figure 4 was not found in Figure 5. Why you can judge whether the transcriptome data is accurate?
Response: I'd like to explain this problem of the reviewer. The genes we selected for qRT-PCR are differentially expressed genes related to anthocyanin synthesis in transcriptome data, which are selected from Table 2. And most of the genes in Figure 5 are not differentially expressed. The qRT-PCR results of these 21 DEGs are consistent with the transcriptome sequencing results, so we can infer that the transcriptome sequencing results are reliable.
Comment 6:"mutant" They used "mutant" in this study in many places.They should clearly indicate "mutant" (PV-PUR?) since the readers are unable to understand which mutant they are saying.
Response: Thank you for your question. We have corrected it according to your suggestions. After "mutant" and "CK", we have clearly marked, and the changes with red revision mode.
Comment 7:The authors must clear the method for the "enrichment of GO terms and KEGG annotations for 209 DEGs"
Response: Thank you for your comments. We have added the description of GO and KEGG enrichment methods in materials and methods, and marked it with red revision mode.
.

Reviewer 3 Report
The article ' Identification and Characterization of a Mutant PV-PUR Gene Responsible for the Purple Phenotype of Snap Bean (Phaseolus vulgaris L.)' presented for review is interesting. However, some issues need to be clarified or supplemented. The comments are included below.
Title
The title is worded correctly and accurately reflects the content.
Abstract
The abstract is clear and adequate.
- Introduction
- The purpose of the research should be clearly stated.
- Results
2.3. The total anthocyanin content of 'pv-pur' is significantly higher than that of 'A18-1'
- Measurement of anthocyanin content - the results were presented with too high accuracy. The spectrophotometric method does not have that high sensitivity.
- Using the phrase that the anthocyanin content was significantly higher suggests that a statistical analysis of the results was performed. This is not confirmed later in the article.
Another
- Please analyze with what precision, how many decimal places should the individual results be presented? Please make the appropriate corrections.
- Materials and Methods
4.3. Determination of total anthocyanins
- Did a single extraction allow complete extraction of the anocyanins? Multiple extractions are required to isolate all anthocyanins. Why, for example, ultrasonically assisted extraction was not used? How do the authors justify the adoption of such a research method?
- Why was the measurement carried out at 530 and 637 nm. Please provide the relevant literature confirming the legitimacy of adopting such a method of conduct.
Another
The statistical analysis of the results was not described in the research methodology.
Conclusion
Conclusions are correct.
Author Response
Dear Reviewer,
We have studied the valuable comments from you carefully, and tried our best to revise the manuscript. The point to point responds to the reviewer’s comments are listed as following:
Comment 1:Introduction- The purpose of the research should be clearly stated.
Response: Thank you for your valuable work. We have made changes to the Introduction, marked with red revision mode.
Comment 2:The total anthocyanin content of 'pv-pur' is significantly higher than that of 'A18-1'- Measurement of anthocyanin content - the results were presented with too high accuracy. The spectrophotometric method does not have that high sensitivity.- Using the phrase that the anthocyanin content was significantly higher suggests that a statistical analysis of the results was performed. This is not confirmed later in the article.Another- Please analyze with what precision, how many decimal places should the individual results be presented? Please make the appropriate corrections.
Response: Thank you for your advice. We do determine the total anthocyanin content with spectrophotometer. Your comment says that the accuracy of our results is too high because we reserve too many decimal places? This is the result of taking the average. Now we change it to keep two decimal places, analyze the difference significance of the results, and change the picture. All corrections are marked in red.
Comment 3: Determination of total anthocyanins- Did a single extraction allow complete extraction of the anocyanins? Multiple extractions are required to isolate all anthocyanins. Why, for example, ultrasonically assisted extraction was not used? How do the authors justify the adoption of such a research method?- Why was the measurement carried out at 530 and 637 nm. Please provide the relevant literature confirming the legitimacy of adopting such a method of conduct.
Response: Thank you for your careful work. We made a mistake in writing here. We wrote 657nm as 637nm, which has been corrected now. In addition, references 31-33 of this method are also added to the article. Corrections and additions are marked in red revision mode.
Comment 4: The statistical analysis of the results was not described in the research methodology.
Response: We added 4.9 ‘Statistical Analysis’ to materials and methods, additions are marked in red revision mode.

Round 2
Reviewer 2 Report
I received and checked some responses from the authors.
Most of them are sufficient for my concerns, however, the response against the comment 2 is not enough at all.
My pointed-out point is to show at least one piece of evidence that supports their conclusion.
But, they changed only words in the text.
Therefore, there is no room to accept the change.
Author Response
Dear Reviewer,
We have studied the valuable comments from you carefully, and tried our best to revise the manuscript.
Comment:I received and checked some responses from the authors.Most of them are sufficient for my concerns, however, the response against the comment 2 is not enough at all.My pointed-out point is to show at least one piece of evidence that supports their conclusion.But, they changed only words in the text.Therefore, there is no room to accept the change.
Response: I'm sorry. Maybe we didn't understand the reviewer's meaning and didn't make correct changes to the second reviewer's comments in the last round. In this revision, we added a subtitle in the conclusion ‘2.8 Prediction and sequence alignment of purple mutant gene PV-PUR’. In this paragraph, we list the inferred basis which phvul 006g018800 (F3'5'h) is the mutant gene causing the mutant purple phenotype. The results of cloning and alignment of the gene sequence in wild-type and mutant were supplemented. The results showed that there were indeed four SNPs in this gene. This is also in line with the characteristics of gene mutation caused by artificial mutation. We hope the reviewers will accept our amendment.

Reviewer 3 Report
The authors made the necessary corrections. The article contains a few editorial errors.
Author Response
Dear Reviewer,
Thank you for your recognition of our work. We have improved some editing problems of the full text according to your comment, and the revised parts are marked with red revision status.
Round 3
Reviewer 2 Report
The authors' response is almost sufficient for my concerns, but they need to improve the following two points:
1)
2.8. Prediction and sequence alignment of purple mutant gene PV-PUR
--->
must be
2.9. Prediction and sequence alignment of purple mutant gene PV-PUR
2) the authors must indicate and explain each mutant cause silent, missense, or nonsense mutation.
Also, they must indicate and discuss the mutations are located in specific domain structure(s) of the protein or not.
Author Response
Dear Reviewer,
We have studied the valuable comments from you carefully, and tried our best to revise the manuscript. The point to point responds to the reviewer’s comments are listed as following:
Comment 1:2.8. Prediction and sequence alignment of purple mutant gene PV-PUR must be 2.9. Prediction and sequence alignment of purple mutant gene PV-PUR
Response: Thank you for your careful work. This is our negligence. We have modified it now.
Comment 2: the authors must indicate and explain each mutant cause silent, missense, or nonsense mutation. Also, they must indicate and discuss the mutations are located in specific domain structure(s) of the protein or not.
Response: Thank you for your valuable work. We added the analysis of SNP and analyzed the changes of amino acids caused by two non-synonymous mutations. The protein domain of the gene was analyzed., the results showed that the first SNP was located in the protein domain. This part has been marked with revision mode. In addition, we supplemented the methods of gene cloning and sequencing in the materials and methods, and labeled them with revision mode.